# Nickel(II)-catalyzed living polymerization of diazoacetates toward polycarbene homopolymer and polythiophene-*block*-polycarbene copolymers

Li Zhou[1,2], Lei Xu[1,2], Xue Song[1], Shu-Ming Kang[1], Na Liu[1] & Zong-Quan Wu [1✉]

Diazoacetate polymerization has attracted considerable research attention because it is an effective approach for fabricating carbon–carbon (C–C) main chain polymers. However, diazoacetate polymerization based on inexpensive catalysts has been a long-standing challenge. Herein, we report a Ni(II) catalyst that can promote the living polymerization of various diazoacetates, yielding well-defined C–C main chain polymers, polycarbenes, with a predictable molecular weight ($M_n$) and low dispersity ($M_w/M_n$). Moreover, the Ni(II)-catalyzed sequential living polymerization of thiophene and diazoacetate monomers affords interesting π-conjugated poly(3-hexylthiophene)-*block*-polycarbene copolymers in high yields with a controlled $M_n$, variable compositions, and low $M_w/M_n$, although the structure and polymerization mechanism of the two monomers differ. Using this strategy, amphiphilic block copolymers comprising hydrophobic poly(3-hexylthiophene) and hydrophilic polycarbene blocks are facilely prepared, which were self-assembled into well-defined supramolecular architectures with tunable photoluminescence.

[1] Department of Polymer Science and Engineering, School of Chemistry and Chemical Engineering, and Anhui Key Laboratory of Advanced Catalytic Materials and Reaction Engineering, Hefei University of Technology, Hefei 230009 Anhui Province, China. [2]These authors contributed equally: Li Zhou, Lei Xu. ✉email: zqwu@hfut.edu.cn

Carbon–carbon (C–C) main chain polymers represent a significant class of polymeric materials that are widely used in daily lives[1,2]. These polymers are typically synthesized via vinyl polymerization through the consecutive addition of C=C bonds, facilitating the construction of the C–C backbone from the two carbon units[3–5]. Diazoacetate polymerization has attracted considerable research attention as an effective approach for fabricating C–C polymers because it is complementary to the aforementioned method[6–10]. The resulting polycarbenes with a polar substituent on each backbone atom are difficult to synthesize using traditional Ziegler–Natta catalysts and other transition metal catalysts[11,12]. Owing to the unique structural characteristics of polycarbenes, they exhibit distinct properties such as enhanced stability, solvent resistance, environmental compatibility, and easy processibility[6–10].

To date, synthetic approaches and catalytic systems for diazoacetate polymerization are very limited. The living polymerization of diazoacetates is a long-standing challenge in the synthetic chemistry community[13–15]. Rhodium and palladium complexes are the most used catalysts for diazoacetate polymerization, developed by de Bruin et al.[7,9,16], Theato et al.[17,18], and Ihara et al.[19–21]. For example, rhodium–diene catalysts can be used to fabricated polycarbenes with a high molecular weight ($M_n$) and tacticity[17,22–25]. Palladium(II)-based catalysts are effective catalysts for various diazoacetate polymerizations and commonly yield atactic polycarbenes with a medium-to-high $M_n$[26–28]. Remarkably, using a Pd(II) catalyst, the polymerization of a unique diazoacetate monomer with a bulky cyclotriphosphazene substituent followed a quasi-living fashion, although the degree of polymerization (DP) was not very high[29]. Very recently, we reported a π-allyl PdCl catalyst with a steric bidentate phosphine ligand, which can polymerize various diazoacetates in a living manner, affording expected polycarbenes in high yields with a controlled $M_n$ and low dispersity ($M_w/M_n$)[13,14]. Thereafter, Toste and coworkers reported a binuclear Pd(II) catalyst that can initiate the quasi living polymerization of diazoacetates[15].

Although significant advances have been achieved in diazoacetate polymerization, the catalysts/initiators used are still limited to expensive palladium and rhodium complexes. Other metal complexes, particularly non-noble metal catalysts, have rarely been reported for diazoacetate polymerization. Therefore, developing inexpensive catalysts/initiators for living diazoacetate polymerization is greatly desired. Nickel catalysts have been widely used in the synthesis of polymers, such as in the controlled radical polymerization of olefins and the catalyst-transfer polymerization (CTP) of aryl monomers for preparing π-conjugated polymers[30,31]. The use of inexpensive nickel complexes for diazoacetate polymerization is also essential. However, the reported nickel-catalyzed diazoacetate polymerization generally affords oil-like oligomers[32,33]. The π-conjugated poly(3-hexylthiophene) (P3HT) is a type of useful semiconductor material and has garnered broad research attention in recent years owing to its intriguing optoelectronic applications[34,35]. Incorporating polycarbenes into P3HT to form a block copolymer can control the self-assembly morphology of P3HT, thereby affording functional materials with great potential in electronics and photonics, light-emitting devices, bioimaging, etc[36–44]. Therefore, the controlled synthesis of P3HT-*block*-polycarbene copolymers is worthy of investigations.

In this work, we report a Ni(II) catalytic system for living diazoacetate polymerization, affording well-defined polycarbenes with predictable $M_n$ and low $M_w/M_n$. Furthermore, functional polymers such as P3HT carrying the Ni(II)-complex terminal can initiate living diazoacetate polymerization, affording π-conjugated block copolymers with a tunable structure and controlled composition. Taking advantage of this method, amphiphilic block copolymers comprising hydrophobic P3HT and hydrophilic polycarbene were readily synthesized, which were self-assembled into well-defined supramolecular architectures with intriguing photoluminescence.

## Results and discussion

**Ni(II)-catalyzed living polymerization of diazoacetates**. A family of Ni(II) complexes with different substituents and phosphine ligands was facilely prepared via the oxidative addition of bis(1,5-cyclooctadiene)nickel(0) to aryl halides or π-allyl trifluoroacetate when different ligands are present in toluene at 25 °C (Fig. 1). The as-prepared catalysts were immediately used in the following polymerization without isolation and characterization to prevent possible decomposition. To investigate the activity of nickel catalysts, benzyl diazoacetate (**1a**) was prepared and used in the polymerization. The polymerizations were performed in tetrahydrofuran (THF) at 25 °C ([**1a**]$_0$ = 0.5 M, [**1a**]$_0$/[Ni(II)]$_0$ = 100). The polymerization performed using π-allyl nickel(II) catalysts could only afford oil-like oligomers, regardless of the ligands (runs 1–6, Table 1), whereas some phenyl nickel(II) complexes (Ph–Ni(L)Br) could afford solid polymers. Size exclusion chromatographic (SEC) analysis performed on the $M_n$ and $M_w/M_n$ of the obtained polymers are summarized in Table 1 and Supplementary Fig. 1. The polymers fabricated using Ph–Ni(L)Br catalysts with appropriate ligands showed a unimodal and symmetrical SEC trace (runs 7–12, Table 1). The $M_n$ and $M_w/M_n$ were ~3 kDa and 1.20, respectively. This study revealed that an aryl nickel(II) complex can be a catalyst for diazoacetate polymerization. Very interestingly, bisthienyl nickel(II) catalyst (BT–Ni(dppp)Cl) with a diphenylphosphine propane (dppp) ligand showed the best result in the polymerization of **1a** among the tested catalysts (runs 13–16 in Table 1 and Supplementary Fig. 2)[45,46]. The recorded SEC peaks of the polymers achieved using BT–Ni(dppp)Cl were symmetric and unimodal, shifting to a high-$M_n$ region with increasing [**1a**]$_0$/[Ni(II)]$_0$ values (Fig. 2a

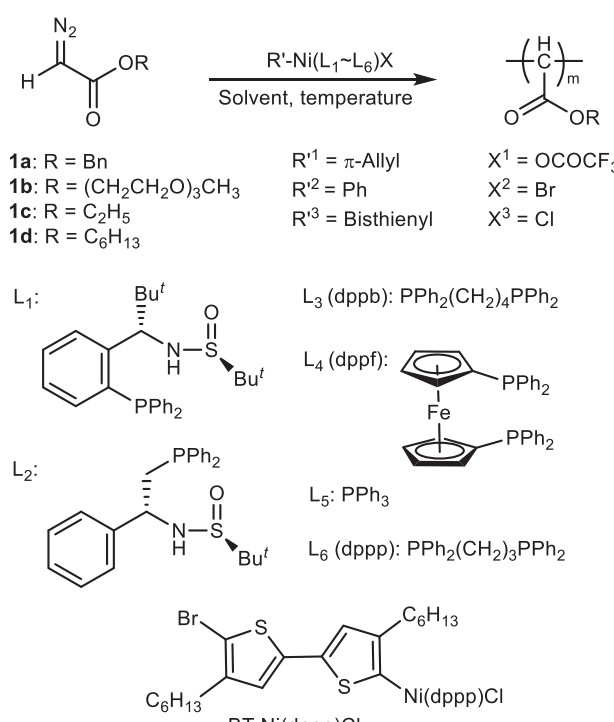

**Fig. 1 Reaction scheme.** Polymerization of monomer **1a–1d** using R′–Ni(L$_1$–L$_6$)X (L = Ligand).

**Table 1 Results for 1a polymerization using various Ni(II) catalysts.**

| run | Catalyst | $[1a]_0/[Ni]_0$ | $M_n{}^a$ (kDa) | $M_w/M_n{}^a$ | polymer | Yield[b] |
|---|---|---|---|---|---|---|
| 1 | π-allyl-Ni(L$_1$)(OCOCF$_3$) | 100 | —[c] | — | Oil-like | 12% |
| 2 | π-allyl-Ni(L$_2$)(OCOCF$_3$) | 100 | —[c] | — | Oil-like | 15% |
| 3 | π-allyl-Ni(L$_3$)(OCOCF$_3$) | 100 | —[c] | — | Oil-like | 24% |
| 4 | π-allyl-Ni(L$_4$)(OCOCF$_3$) | 100 | —[c] | — | Oil-like | 18% |
| 5 | π-allyl-Ni(L$_5$)(OCOCF$_3$) | 100 | —[c] | — | Oil-like | 15% |
| 6 | π-allyl-Ni(L$_6$)(OCOCF$_3$) | 100 | —[c] | — | Oil-like | 30% |
| 7 | Ph-Ni(L$_1$)Br | 100 | —[c] | — | Oil-like | 12% |
| 8 | Ph-Ni(L$_2$)Br | 100 | —[c] | — | Oil-like | 20% |
| 9 | Ph-Ni(dppb)Br | 100 | 2.83 | 1.16 | Solid | 30% |
| 10 | Ph-Ni(dppf)Br | 100 | —[c] | — | Oil-like | 32% |
| 11 | Ph-Ni(PPh$_3$)Br | 100 | 2.83 | 1.20 | Solid | 15% |
| 12 | Ph-Ni(dppp)Br | 100 | 3.46 | 1.15 | Solid | 20% |
| 13 | BT-Ni(dppp)Cl | 50 | 5.12 | 1.17 | Solid | 65% |
| 14 | BT-Ni(dppp)Cl | 100 | 10.78 | 1.18 | Solid | 68% |
| 15 | BT-Ni(dppp)Cl | 150 | 14.80 | 1.19 | Solid | 72% |
| 16 | BT-Ni(dppp)Cl | 200 | 19.83 | 1.20 | Solid | 78% |

[a]$M_n$ and $M_w/M_n$ values were determined by SEC using polystyrene standards.
[b]The isolated yields.
[c]$M_n$ was lower than the detection limit of SEC.

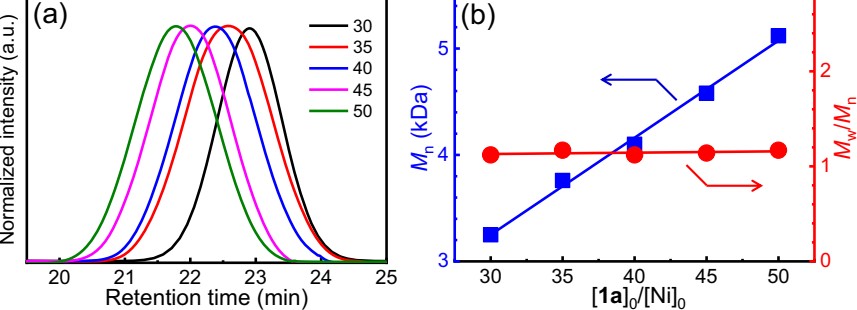

**Fig. 2 SEC traces and result analyses. a** SEC traces of poly-$1a_m$s prepared using BT-Ni(dppp)Cl-catalyzed **1a** polymerization in different $[1a]_0/[Ni]_0$ ratios in THF at 25 °C. **b** Plots of $M_n$ and $M_w/M_n$ of the poly-$1a_m$ vs. **1a**-to-Ni(II) ratio.

and runs 1–4 in Supplementary Table 1). Remarkably, $M_n$ exhibited a linear correlation with the feed ratio of **1a** to the Ni(II) catalyst, and all yielded polymers exhibited a low $M_w/M_n$ of <1.20 (Fig. 2b). These results demonstrate that BT-Ni(dppp)Cl is an active catalyst for diazoacetate polymerization and may follow a living polymerization mechanism. Notably, the polymerization yield was not very high because the polymers showed good solubility in common solvents, resulting in polymer losses during the isolation process. Next, the polymerization of **1a** was performed using BT-Ni(dppp)Cl in different solvents. Interestingly, all polymerizations yielded expected polymers with a controlled $M_n$ and low $M_w/M_n$, except for those performed in chloroform (runs 5–7 in Supplementary Table 1), probably because of the weak acidity of chloroform (run 8 in Supplementary Table 1). Furthermore, the temperature exerted only a slight effect on the polymerization (runs 9 and 10 in Supplementary Table 1). Thus, performing polymerization in THF at room temperature is the ideal condition. Because BT-Ni(dppp)Cl showed good performance in polymerization, it was subjected to careful isolation and complete characterization (Supplementary Figs. 3–10)[47].

To obtain polymerization details, **1a** was polymerized using BT-Ni(dppp)Cl by employing polystyrene (PSt; $M_n = 41.4$ kDa and $M_w/M_n = 1.02$) as the internal standard ($[1a]_0/[Ni]_0 = 100$). The polymerization was monitored using SEC to determine the monomer conversion as well as the $M_n$ and $M_w/M_n$ data of the yielded polymers. As expected, the yielded polymers exhibited unimodal elution peaks in the SEC trace and continually shifted

to high-$M_n$ regions during the polymerization process (Fig. 3a). More than 85% of **1a** was polymerized within 5 h (Fig. 3b). The plot of $-\text{Ln}([M]/[M]_0)$ as a function of the polymerization time revealed that the polymerization followed the first-order reaction mechanism, and the rate constant was $8.3 \times 10^{-5}$ s$^{-1}$. Furthermore, the $M_n$ of the synthesized polymers was proportionally and linearly correlated with the **1a** conversion; all fabricated polymers showed a low $M_w/M_n$ of <1.20 (Fig. 3c). To further prove the living nature of the polymerization, a new feed of **1a** was added to the polymerization solution of the freshly prepared poly-$1a_{50}$ ($M_n = 5.52$ kDa and $M_w/M_n = 1.19$) in THF at 25 °C ($[1a]_0/[Ni]_0 = 30$). SEC analyses verified the occurrence of the chain extension because the SEC peak moved to high-$M_n$ regions, maintaining a unimodal elution trace (Supplementary Fig. 11). The $M_n$ increased to 8.47 kDa with $M_w/M_n = 1.20$. Collectively, these findings prove the living nature of the diazoacetate polymerization initiated using the BT-Ni(dppp)Cl catalyst.

**Structural characterization.** The as-prepared poly-$1a_m$ was verified using $^1$H nuclear magnetic resonance ($^1$H NMR), $^{13}$C nuclear magnetic resonance ($^{13}$C NMR), Fourier transform infrared spectroscopy (FT-IR), and mass spectrometry. Characteristic resonances can be observed in the $^1$H NMR trace of poly-$1a_{20}$ (Fig. 4a). The signals at 7.43–6.85 (*e*) and 5.11–4.18 (*d*) ppm were assigned to benzene and benzyl methylene protons,

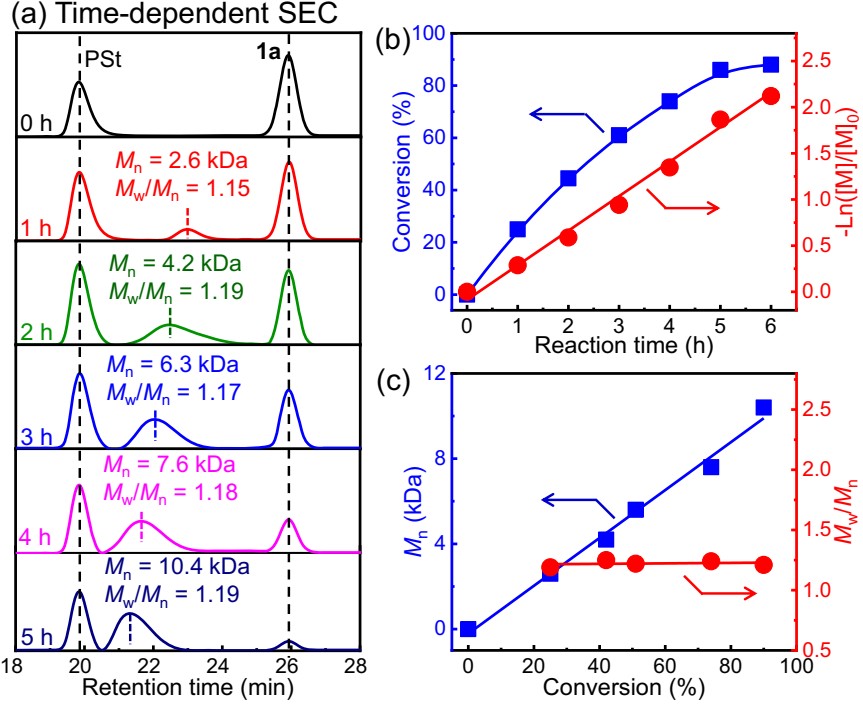

**Fig. 3 Time-dependent SEC analyses. a** SEC for BT–Ni(dppp)Cl-catalyzed **1a** polymerization using polystyrene (PSt) as the internal standard (THF, 25 °C, [**1a**]$_0$/[Ni]$_0$ = 100). **b** Plots of **1a** conversion and –Ln([M]/[M]$_0$) values vs. polymerization time. **c** Plots of the $M_n$ and $M_w/M_n$ of the fabricated polymers against **1a** conversion.

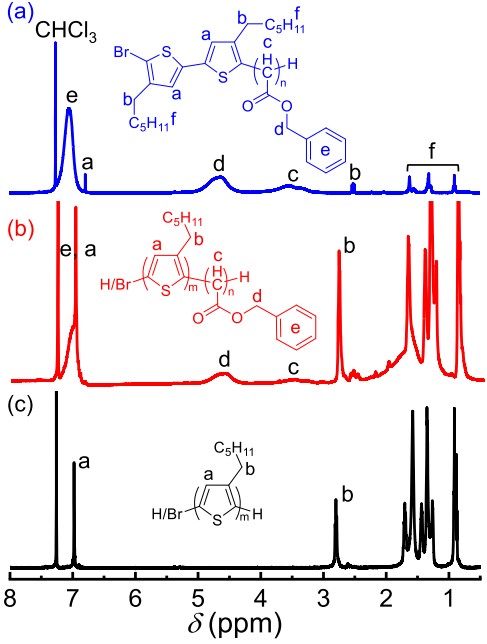

**Fig. 4 $^1$H NMR spectra.** $^1$H NMR (400 MHz, CDCl$_3$) spectra of poly-**1a**$_{20}$ (**a**), poly(**2**$_{20}$-b-**1a**$_{40}$) (**b**), and poly-**2**$_{20}$ (**c**) obtained at room temperature.

respectively. Further, the broad and weak resonances at 3.88–3.02 ppm (*c*) corresponded to the main chain CH. Moreover, signals attributed to the ArH and ArCH$_2$ of the terminal BT unit were observed at 6.81 (*a*) and 2.51 (*b*) ppm, respectively, although they were weak. The DP deduced from the resonances at 6.81 ppm (*a*, ArH of the terminal BT unit) and 3.88–3.02 ppm (*c*, CH of the main chain) was ~20 for poly-**1a**$_{20}$, generally agreeing with the expected structure. The $^{13}$C NMR trace revealed characteristic resonances at 170.2, 66.6, and 46.1 ppm, assigned to the carbonyl

and OCH$_2$ carbons of the pendants and the CH carbon of the backbone, respectively (Supplementary Fig. 12). The FT-IR spectrum of poly-**1a**$_{50}$ also verified the formation of the polymer, observed based on the absorption at 1730 cm$^{-1}$ attributed to the C=O vibration (Supplementary Fig. 13). The poly-**1a**$_{20}$ structure was further verified by matrix-assisted laser desorption/ionization spectroscopy equipped with time-of-flight detection mass spectroscopy, which showed that the highest peak of the 21-mer polymer with the BT unit was present at the living chain end (Supplementary Fig. 14). The difference between the adjacent peaks of the mass spectrum was ~148.1 g/mol, agreeing well with the $M_n$ of the repeating unit. These analyses confirm the formation of the polymers.

Using the BT–Ni(dppp)Cl catalyst, a family of poly-**1a**$_m$s with a predictable $M_n$ and low $M_w/M_n$ was facilely synthesized (runs 13–16 in Table 1 and runs 1–4 in Supplementary Table 2). In addition to **1a**, the BT–Ni(dppp)Cl catalyst could catalyze the living polymerization of monomer **1b** with a hydrophilic triethylene glycol monomethyl ether chain and **1c** and **1d** with alkyl chains of different lengths. All polymerizations afforded desired polymers with the expected $M_n$ and low $M_w/M_n$ (runs 11–14 in Supplementary Table 1 and Supplementary Figs. S15–S23), further confirming the high polymerization activity of the BT–Ni(dppp)Cl catalyst.

**Synthesis and characterization of P3HT-b-polycarbene copolymers.** Because BT–Ni(dppp)Cl showed excellent performance in diazoacetate polymerization, functional polymers with a similar Ni(II)-complex terminal may also initiate the living polymerization of diazoacetate, consequently yielding interesting block copolymers. To verify this assumption, 2-bromo-3-hexyl-5-chloromagnesiothiophene (**2**) obtained from 2-bromo-3-hexyl-5-iodothiophene was polymerized in-situ using Ni(dppp)Cl$_2$ in THF at 25 °C ([**1a**]$_0$ = 0.50 M, [**1a**]$_0$/[Ni(II)]$_0$ = 20), followed by the CTP or Grignard metathesis mechanism (Fig. 5)[38,45–50].

The achieved poly-$2_{20}$ showed a $M_n$ of 6.71 kDa with $M_w/M_n =$ 1.19 (Fig. 6a). The polymerization solution was directly treated with a solution of **1a** in THF under dry nitrogen ([**1a**]$_0$/[Ni]$_0$ = 40). After stirring at room temperature for 24 h, the solution was precipitated into diethyl ether. A P3HT-*b*-polycarbene copolymer poly($2_{20}$-*b*-$1a_{40}$) was isolated in 71% yield via centrifugation.

The recorded SEC trace of poly($2_{20}$-*b*-$1a_{40}$) showed a movement to the shorter elution time area compared with that of poly-$2_{20}$ macroinitiator (Fig. 6a); and maintained a unimodal and symmetric peak. The $M_n$ of poly($2_{20}$-*b*-$1a_{40}$) was 12.63 kDa, higher than that of poly-$2_{20}$ ($M_n$ = 6.71 kDa and $M_w/M_n$ = 1.19), while the $M_w/M_n$ was as low as 1.18, indicating that the one-pot hybrid block copolymerization probably proceeded in the living polymerization mechanism. To confirm this hypothesis, a range of block copolymerizations was performed by adding identical amounts of **1a** with different amounts of the Ni(II)-terminated poly-$2_{20}$ macroinitiator. The correlations between the $M_n$ and $M_w/M_n$ of the produced block copolymers with [**1a**]$_0$/[Ni(II)]$_0$ ratios were plotted in Fig. 6b. As expected, a linear correlation between $M_n$ and [**1a**]$_0$/[Ni(II)]$_0$ was observed and all yielded block copolymers showed a low $M_w/M_n$ of <1.20 (Supplementary Fig. 24). This result demonstrates that the block copolymerization of diazoacetates with Ni(II)-terminated P3HT follows a living polymerization mechanism.

The $^1$H NMR trace of poly($2_{20}$-*b*-$1a_{40}$) showed signals corresponding to both poly-$2_m$ and poly-$1a_n$ blocks (Fig. 4b and c). For instance, the resonances of ArH and ArCH$_2$ for the poly-$2_{20}$ segment at 6.91 (*a*) and 2.81–2.61 (*b*) ppm were evident (Fig. 4b). Moreover, the resonance of CO$_2$CH$_2$ for the poly-$1a_m$ block at 5.11–4.18 ppm (*d*) could be detected. Based on the integral analyses of ArCH$_2$ (*b*) and CO$_2$CH$_2$ (*d*) from the poly-$2_m$ and poly-$1a_m$ blocks, respectively, the ratio of the two blocks was approximately 1:2 for poly($2_{20}$-*b*-$1a_{40}$), generally consistent with the feed ratio of the used monomers. The FT-IR spectra also verified the poly($2_{20}$-*b*-$1a_{40}$) structure based on the detection of the characteristic vibrations of the two blocks (Supplementary Fig. 25). Collectively, these analyses prove the successful synthesis of the desired P3HT-*b*-polycarbene copolymer and support the one-pot hybrid block copolymerization of thiophene and diazoacetate monomers that follow the living polymerization mechanism, even though the structure and polymerization mechanism of the two monomers differ. Using this copolymerization approach, a family of P3HT-*b*-polycarbene copolymers with the desired $M_n$ and low $M_w/M_n$ was readily synthesized (runs 1–7 in Table 2 and Supplementary Fig. 26). Moreover, interesting amphiphilic π-conjugated block copolymer poly($2_m$-*b*-$1b_n$)s were facilely obtained using diazoacetate **1b** as the second monomer (runs 8–10 in Table 2 and Supplementary Figs. 27–29). Note that an attempt was made to prepare the block copolymer poly($2_m$-*b*-$1a_n$) under a reverse sequence by treating the freshly prepared poly-$1a_{50}$ ($M_n$ = 5.03 kDa and $M_w/M_n$ = 1.18) with monomer **2** ([**2**]$_0$ = 0.2 M, [**2**]$_0$/[Ni] = 20). However, no chain extension occurred, as revealed by SEC analysis, even the copolymerization was performed at 55 °C for 48 h. The reason is not very clear; the Grignard reagent **2** possibly preferred to react with the ester pendants of poly-$1a_m$, preventing the chain extension.

**Self-assembly of poly($2_{20}$-*b*-$1b_{40}$) copolymer.** The self-assembly behavior of the amphiphilic poly($2_{20}$-*b*-$1b_{40}$) copolymer was investigated in different solvents. Because of the amphiphilic nature of this copolymer, it can be dissolved in many solvents such as CHCl$_3$, THF, toluene, methanol, and even water. However, the absorption and emission behaviors of this copolymer differ in solvents. Poly($2_{20}$-*b*-$1b_{40}$) exhibited a yellow color in THF and a purple–red color in isopropanol (IPA) (Fig. 7a and Supplementary Fig. 30). The ultraviolet–visible (UV–vis) absorption spectra of poly($2_{20}$-*b*-$1b_{40}$) in the THF/IPA mixture are shown in Fig. 7a. In pure THF, poly($2_{20}$-*b*-$1b_{40}$) showed a large absorption in long-wavelength regions, attributed to the P3HT block with the maximum absorption located at 445 nm, similar to the absorption of the P3HT homopolymer. Owing to the absence of the π-conjugated system, poly-$1b_{40}$ showed almost no absorption in the long-wavelength region[13]. When IPA, a polar solvent, was added, the absorption of poly($2_{20}$-*b*-$1b_{40}$) at 445 nm decreased. Further, three new absorption peaks appeared at 511, 553, and 604 nm, indicating the π–π stacking and crystalline aggregation of the P3HT block (Fig. 7a)[35–37]. The emission of the block copolymer changed in accordance with the absorption change. Poly($2_{20}$-*b*-$1b_{40}$) showed a yellow emission of the P3HT block under illumination at 365 nm in THF, whereas it showed a blue emission in IPA under the same conditions (Fig. 7b and Supplementary Fig. 30). This change was further confirmed by photoluminescent (PL) spectra. When excited at

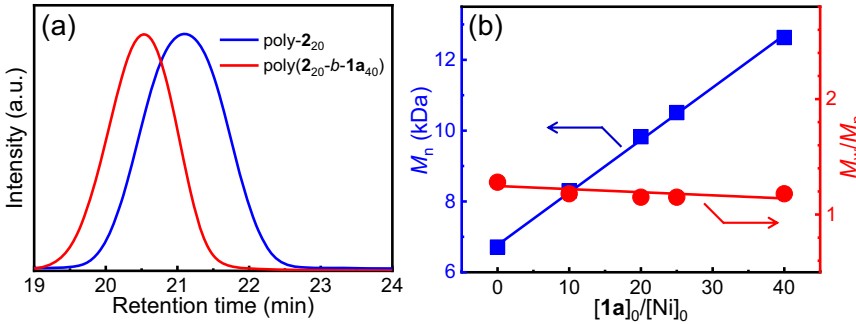

**Fig. 5 Synthesis of P3HT-*b*-polycarbene copolymer.** Synthetic route for poly($2_m$-*b*-$1a_n$) and poly($2_m$-*b*-$1b_n$) copolymers.

**Fig. 6 SEC analyses of block copolymerization. a** SEC traces of poly-$2_{20}$ and the corresponding poly($2_{20}$-*b*-$1a_{40}$) copolymer. **b** Plots of the $M_n$ and $M_w/M_n$ data of poly($2_{20}$-*b*-$1a_n$)s *vs.* [**1a**]$_0$/[Ni]$_0$ ratio for the copolymerization of **1a** initiated by poly-$2_{20}$ with the Ni(II) complex at the chain end.

**Table 2 Characterization for P3HT-*b*-polycarbene copolymers.**

| Run | Polymer | poly-2$_m$[a] | | Block polymer | | |
|-----|---------|-----------------|-------------|------------------|-------------|--------|
| | | $M_n$[b] (kDa) | $M_w/M_n$[b] | $M_n$[b] (kDa) | $M_w/M_n$[b] | Yield[c] |
| 1 | poly(**2**$_{20}$-*b*-**1a**$_{40}$) | 6.71 | 1.19 | 12.63 | 1.18 | 71% |
| 2 | poly(**2**$_{20}$-*b*-**1a**$_{30}$) | 6.71 | 1.19 | 10.51 | 1.15 | 65% |
| 3 | poly(**2**$_{20}$-*b*-**1a**$_{25}$) | 6.71 | 1.19 | 9.83 | 1.15 | 60% |
| 4 | poly(**2**$_{18}$-*b*-**1a**$_{15}$) | 5.86 | 1.20 | 6.58 | 1.19 | 62% |
| 5 | poly(**2**$_{13}$-*b*-**1a**$_{10}$) | 4.75 | 1.18 | 5.87 | 1.16 | 68% |
| 6 | poly(**2**$_{18}$-*b*-**1a**$_{20}$) | 5.90 | 1.18 | 8.17 | 1.16 | 62% |
| 7 | poly(**2**$_{18}$-*b*-**1a**$_{25}$) | 5.90 | 1.18 | 9.17 | 1.17 | 63% |
| 8 | poly(**2**$_{20}$-*b*-**1b**$_{40}$) | 6.71 | 1.19 | 10.12 | 1.14 | 72% |
| 9 | poly(**2**$_{20}$-*b*-**1b**$_{20}$) | 6.71 | 1.19 | 8.25 | 1.18 | 68% |
| 10 | poly(**2**$_{15}$-*b*-**1b**$_{20}$) | 4.83 | 1.18 | 7.13 | 1.19 | 62% |

[a]$M_n$ and $M_w/M_n$ were obtained by SEC just before the addition of the second monomer.
[b]$M_n$ and $M_w/M_n$ were obtained from SEC analyses.
[c]The isolated yields over two steps.

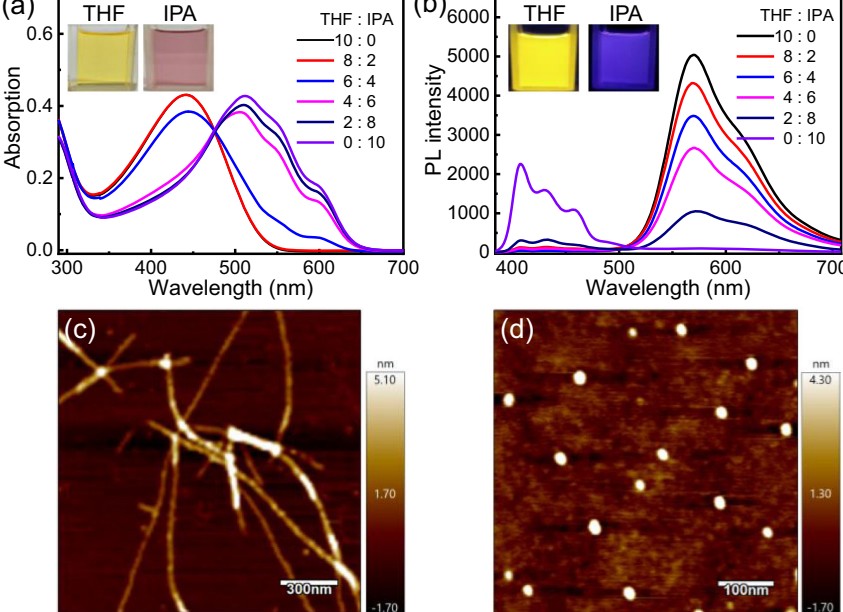

**Fig. 7 Absorption and emission spectra as well as atomic force microscopic (AFM) images of poly(2$_{20}$-*b*-1b$_{40}$).** Absorption (**a**) and emission (**b**) spectra of poly(**2**$_{20}$-*b*-**1b**$_{40}$) in the THF/IPA mixture ($c = 0.3$ mg/mL and $\lambda_{exc} = 365$ nm). Insets show images of poly(**2**$_{20}$-*b*-**1b**$_{40}$) in IPA and THF under room light (**a**) and UV light at 365 nm (**b**). AFM height images of poly(**2**$_{20}$-*b*-**1b**$_{40}$) cast from THF (**c**) and IPA (**d**) solutions onto silicon wafers.

365 nm, poly(**2**$_{20}$-*b*-**1b**$_{40}$) showed weak and strong emissions in short- (400–500 nm) and long-wavelength (500–700 nm) regions, respectively, with the maximum emission located at ~570 nm (Fig. 7b). The emission in the long-wavelength region was gradually quenched after the addition of IPA, whereas that in the short-wavelength region increased. In pure IPA, the block copolymer showed emissions only in the short-wavelength region, with the maximum emission located at ~420 nm, thus exhibiting a blue emission. The solvent-induced optical changes were attributed to the solvophobic effect of the amphiphilic copolymer in selective solvents, affording distinct supramolecular architectures[51,52].

The self-assembled morphologies of the block copolymer were studied using atomic force microscopy (AFM). Poly(**2**$_{20}$-*b*-**1b**$_{40}$) was self-assembled into nanofibers in THF with a diameter of ~60 nm and a constant length of up to several micrometers (Fig. 7c), whereas it was self-assembled into spherical nanoparticles in IPA with a diameter of ~30 nm (Fig. 7d). These results suggest that the absorption and emission changes were probably attributed to different self-assembled architectures, consistent with the findings reported in the literature[13,51,52]. In nonselective THF, the block copolymer showed the emission and absorption of the P3HT block. However, in the presence of IPA, the block polymer self-assembled into compacted spherical nanoparticles. The π-conjugated structure of the P3TH block was altered during the self-assembly process; thus, the block copolymer exhibited different absorption and emission properties[35,51]. The self-assembled morphology of poly(**2**$_{20}$-*b*-**1b**$_{40}$) in IPA was further confirmed using transmission electron microscopy (TEM) (Supplementary Fig. 31). As expected, the TEM image of the sample cast from poly(**2**$_{20}$-*b*-**1b**$_{40}$) in dilute IPA showed clear core–shell-like spherical nanoparticles with good homogeneity. The average diameter of these nanoparticles was ~32 nm, consistent with AFM investigations. Dynamic light scattering analyses revealed that the hydrodynamic diameter of poly(**2**$_{20}$-*b*-**1b**$_{40}$) was 33.7 nm in IPA, even at a low concentration of the polymer of 0.3 mg/mL, indicating the self-assembly of the block copolymer in IPA (Supplementary Fig. 32). Note that the

**Fig. 8 Proposed polymerization mechanism. a** Proposed mechanism for the Ni(II)-mediated diazoacetate polymerization of **1a**, and preliminary density functional theory (DFT) calculations for the initiating reaction of **1a** using Ph–Ni(dppp)Br (**b**) and BT–Ni(dppp)Cl (**c**).

polycarbene block comprised C–C bonds and showed densely packed substituents on every backbone atom, rendering it with high rigidity and self-assembly tendency, thereby imparting the P3HT-*b*-polycarbene copolymer with distinct self-assembly-induced PL[18,21,24].

Lastly, the mechanism of Ni(II)-catalyzed living diazoacetate polymerization was proposed[27]. The initiation process was triggered by an attack of an $\alpha$-carbon atom of diazoacetate on the nickel(II) center (Fig. 8). Furthermore, the transfer of Ar in the nickel catalyst onto the $\alpha$-carbon was accompanied by the release of $N_2$. Based on this reaction, the monomer was inserted into the Ar–Ni(II) catalyst, affording an intermediate (II). Subsequently, catalyzed using the nickel-complex of II, a new diazoacetate monomer released $N_2$ and the generated carbine was inserted into the C-Ni(II), yielding an intermediate III. By repeating the insertion reaction, the polymer chain was propagated. Based on this mechanism, the Ar group in the Ni(II) catalyst was transferred onto the initiating chain end of the generated polymers. Thus, the Ni(II)-terminated P3HT–catalyzed the living polymerization of diazoacetates afforded expected P3HT-*b*-polycarbene copolymers.

This mechanism was supported using preliminary density functional theory (DFT) calculations (Fig. 8b and c, Supplementary Table 2, and Supplementary Figs. 33–37), which also reveal the different activity of phenyl Ni(II) catalyst (Ph–Ni(dppp)Br) with BT-Ni(dppp)Cl. Ph–Ni(dppp)Br (**Ph-IN1**, 0.0 kcal/mol, reference point) coordinated with **1a** via the transition of **Ph-TS1** (28.7 kcal/mol) to **Ph-IN2** (25.6 kcal/mol) (Fig. 8b). Subsequently, based on a concerted transition state **Ph-TS2** (38.9 kcal/mol), in which the occurrence of the C–N bond cleavage and formation of the C–C bond were observed (Supplementary Fig. 33), an intermediate **Ph-IN3** (−45.6 kcal/mol) was generated.

After releasing $N_2$, Br atom went back to the *trans*-position of the Ni(II) atom in the P–Ni–P plane, affording **Ph-IN4** (−51.5 kcal/mol). Obviously, the rate-determining step was the concerted C–N bond cleavage and C–C bond formation processes via **Ph-TS2**. The reaction of BT–Ni(dppp)Cl with **1a** was also explored using DFT calculations (Fig. 8c). Unlike in the case of Ph–Ni(dppp)Br, only the key rate-determining step (**BT-TS2**, 38.2 kcal/mol), reactant (**BT-IN1**, 0.0 kcal/mol), and product **BT-IN4** (−47.0 kcal/mol) were obtained, suggesting an irreversible reaction pathway. In **BT-TS2**, the C–N cleavage also occurred concertedly with the formation of the C–C bond (Supplementary Fig. 34). By comparing the reaction pathways between Ph–Ni(dppp)Br and BT–Ni(dppp)Cl, **1a** was found to be was more favorable to react with Ph–Ni(dppp)Br (**Ph-TS2**) than with BT–Ni(dppp)Cl (**BT-TS2**). The ground state of **Ph-IN4** (IIa in Fig. 8a) formed more easily and was more stable than **BT-IN4** (IIb in Fig. 8a); therefore, the chain extension of **1a** using **Ph-IN4** was more difficult than that using **BT-IN4**. Consequently, the expected polycarbenes were obtained with a higher yield and a predictable $M_n$ using the BT–Ni(dppp)Cl catalyst, whereas polymers with a lower yield and $M_n$ were afforded using the Ph–Ni(dppp)Br catalyst under identical experimental conditions. Note that the influence of the Ph and BT units on the polymerization would decrease considerably with the chain extension because the distance between these units and the living chain end increases.

In conclusion, we developed a Ni(II) catalyst that can promote the living polymerization of diazoacetates and yield C–C main chain polymers with a desired $M_n$ and low $M_w/M_n$. Furthermore, P3HT with a Ni(II) complex at the chain end can initiate the living polymerization of diazoacetates, yielding well-defined P3HT-*b*-polycarbene copolymers with a tunable $M_n$ and narrow

$M_w/M_n$. An amphiphilic block copolymer prepared using this method could be self-assembled into distinct nanostructures in selective solvents with unique optical properties. The findings of this study will provide a strategy for the controlled synthesis of C–C main chain polymers with polar functionality on every backbone atom and propose an approach for the controlled synthesis of semiconductor materials with great potential in various applications, such as optoelectronics, semiconductor devices, and bioimaging.

## Data availability

Synthetic details and additional experimental data generated in this study are provided in the Supplementary Information.

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

## Acknowledgements

This work is supported by the National Natural Science Foundation of China (NSFC, Grant Nos. 21971052, N.L.; 22071041, Z.W.; 21871073, Z.W.; and No. 51903072, L.Z.). Z.W. and L.Z. thank the Fundamental Research Funds for the Central Universities of China (Grant Nos. PA2019GDPK0057, L.Z.; and PA2020GDJQ0028, Z.W.). L.X. thanks the financial support from the China Postdoctoral Science Foundation (Grant No. 2020M681981, L.X.).

## Author contributions

Z.-Q.W. and N.L. designed and directed the project; L.Z., L.X., and X.S. performed the experiments. L.X. performed the AFM experiments. S.-M.K. performed DFT calculations. Z.-Q.W. and L.Z. wrote the manuscript with input from all other authors.

## Competing interests

The authors declare no competing interests.
