## [Peer Review File · Nature Communications]

Nickel(II)-Catalyzed Living Polymerization of Diazoacetates toward Polycarbene Homopolymer and Polythiophene-block-Polycarbene CopolymersREVIEWER COMMENTS

Reviewer #1 (Remarks to the Author):

This paper by Wu and coworkers describes the quasi-living polymerization of diazoacetates catalyzed by Ni(II) complexes. Block copolymerization with polythiophenes was also demonstrated. The paper has several interesting aspects, in particular the use of Ni(II) for these reactions. The block copolymers show solvato-chromic behavior pointing to pi-stacking and (partial) phase separation of the blocks in solution depending of the solvent mixtures used. I have reviewed an earlier submission of this paper to another journal, for which I recommended some changes. These changes were made, and have substantially improved the paper. Overall I think this paper is suitable for publication now. However, one issue was not addressed in this new submission, so I repeat that recommendation here.

- An important novelty is that the carbene-polymerization reaction can be performed with new Ni(II) catalysts. As such, for publication in JACS, these new catalysts need to be fully characterized. How else can readers couple these finding to the new structures?

Reviewer #2 (Remarks to the Author):

The manuscript describes a Ni-catalyzed living polymerization of diazo-compounds. The described method was also applied to synthesize block copolymer containing conjugated blocks. The morphology of copolymers cast from different solvents was also examined. Overall, the work can potentially be suitable for Nature Communication; however, significant improvement to the writing and structure of the manuscript is required. Additionally, in order to improve the impact and scholarship of the work, additional experiments and understanding of the observed absorption/emission behavior must be provided. Some specific comments:

(1). As stated above, at points the writing makes the manuscript almost unreadable. Here as some specific examples:

- This sentence does not make sense: "In which C-C was formed via sequential carbene insertions derived from diazoacetate monomers after N₂ release."

- "...polymerization is of great desired." ...polymerization is highly desirable?

- "...can be retrospect to 1970s'. ???

- "...FT-IR, and mass spectra." ...FT-IR and mass spectrometry?

There are many, many more examples where the writing makes it difficult to understand.

(2). What does "the first reaction rate law" mean? I assume the authors mean a first order reaction?

(3). The authors state "These results suggested the absorption and emission changes were probably from the different self-assembled architectures." This seems unlikely given the low concentration of the samples in absorption/emission compared to the AFM samples which contain no solvent. The authors need to support this claim, either experimentally or with literature examples.

(4) The authors state "with the presence of selective solvent IPA, the polymer self-assembled into compacted spherical nanoparticles."

What is the evidence for this? For example, can this be observed by DOSY NMR?

(5). In Table 1, entries 1-6 do not have M_n data. Why? Was it lower than the detection limit?

(6). The sentence beginning with "Nickel catalysts have been widely used in polymer synthesis..." needs some citations to examples of Nickel catalysts in polymer synthesis to support the authors statement that they have been widely used.

-

Reviewer #3 (Remarks to the Author):

This is an interesting paper that describes a novel living polymerization of diazoacetates with a non-expensive Ni(II) catalyst and further living block polymerization of 3-hexylthiophene, resulting in photoluminescence block copolymers. Poly(acylmethylene)s, an interesting class of polymers that are different from commodity plastics, polyolefins, have been extensively studied during the past decade. Nowadays Pd(II)-based catalysts developed by the group of Wu and others enable to polymerize diazoacetates in a living fashion to produce the poly(acylmethylene)s with a controlled molar mass and a narrow molar mass distribution. The main advance in this paper lies in the discovery of a unique bithienyl Ni(II) catalyst carrying a dppp ligand (BT-Ni(dppp)Cl), which promoted the living homo- and block polymerizations. The development of non-expensive, non-noble metal, Ni-based catalysts/initiators for living polymerization of diazoacetate appears to be a great achievement, and the results are worthy of publication.

The work has been carried out to a high standard and the manuscript is well written based on the experimental synthesis results. It is not clear, however, whether the work has the exceptional level of significance required for publication in this journal. Perhaps, the mechanism of the living polymerization of diazoacetates initiated and propagated by the BT-Ni(dppp)Cl should be proposed/discussed. The rationale for the reactivity of BT-Ni(dppp)Cl that is better than PhNi(dppp)Br should be also discussed; DFT calculations using crystal structures of BT-Ni(dppp)Cl and PhNi(dppp)Br may be useful. In addition, the authors should stress the superiority of the present photoluminescence block copolymers over other PT-based block copolymers together with physical properties of novel homo- and block copolymers. Otherwise, publication in a more specialized journal might be appropriate.

Reviewer #1:

Comments: This paper by Wu and coworkers describes the quasi-living polymerization of diazoacetates catalyzed by Ni(II) complexes. Block copolymerization with polythiophenes was also demonstrated. The paper has several interesting aspects, in particular the use of Ni(II) for these reactions. The block copolymers show solvent chromic behavior pointing to pi-stacking and (partial) phase separation of the blocks in solution depending of the solvent mixtures used. I have reviewed an earlier submission of this paper to another journal, for which I recommended some changes. These changes were made, and have substantially improved the paper. Overall I think this paper is suitable for publication now. However, one issue was not addressed in this new submission, so I repeat that recommendation here.

- An important novelty is that the carbene-polymerization reaction can be performed with new Ni(II) catalysts. As such, for publication in JACS, these new catalysts need to be fully characterized. How else can readers couple these finding to the new structures?

Response: According to the reviewer's suggestion, the catalyst was carefully isolated and fully characterized. (1) The isolated BT-Ni(dppp)Cl was characterized by ^1H NMR, ^{31}P NMR, ^{13}C NMR, mass spectrum, FT-IR, and elemental analysis. The data were provided in SI (page S6 in SI and Supplementary Fig. 3-7). (2) To further confirm the structure, BT-Ni(dppp)Cl was quenched by aqueous hydrochloric acid and the resulting thiophene dimer (BT-H) was verified by ^1H NMR, ^{13}C NMR, FT-IR, and elemental analysis (page S6 in SI and Supplementary Fig. 8-10). We have added a description in the manuscript to address this issue (page 5, line 15-16): "Because BT-Ni(dppp)Cl showed good performance on the polymerization, it was carefully isolated and fully characterized (Supplementary Fig. 3-10).⁴⁷"

Reviewer #2:

Comments: The manuscript describes a Ni-catalyzed living polymerization of diazo-compounds. The described method was also applied to synthesize block copolymer containing conjugated blocks. The morphology of copolymers cast from different solvents was also examined. Overall, the work can potentially be suitable for Nature Communication; however, significant improvement to the writing and structure of the manuscript is required. Additionally, in order to improve the impact and scholarship of the work, additional experiments and understanding of the observed absorption/emission behavior must be provided. Some specific comments:

(1). As stated above, at points the writing makes the manuscript almost unreadable.

Here are some specific examples:

- This sentence does not make sense: "In which C-C was formed via sequential carbene insertions derived from diazoacetate monomers after N₂ release."

- "...polymerization is of great desired." ...polymerization is highly desirable?

- "...can be retrospect to 1970s'. ???

- "...FT-IR, and mass spectra." ...FT-IR and mass spectrometry?

There are many, many more examples where the writing makes it difficult to understand.

Response: According to the reviewer's suggestion, we have corrected the writing mistakes (page 3, line 2 and line 6-8; page 8, line 2-3). Moreover, the writing of this manuscript was corrected by a native English-speaking expert.

(2). What does "the first reaction rate law" mean? I assume the authors mean a first order reaction?

Response: Yes, it is "a first order reaction". We have corrected expression in the manuscript (page 7, line 1).

(3). The authors state "These results suggested the absorption and emission changes were probably from the different self-assembled architectures." This seems unlikely given the low concentration of the samples in absorption/emission compared to the AFM samples which contain no solvent. The authors need to support this claim, either experimentally or with literature examples.

Response: According to the reviewer's suggestion, related papers were cited in ref. 13, 51, and 52 (*J. Am. Chem. Soc.* 2018, 17773–17781; *Macromolecules* 2016, 110–119; *Macromolecules* 2016, 1180–1190) in the manuscript to support this statement (page 14, line 8-10). The samples for AFM investigation were performed by casting the dilute solutions of the block copolymer ($c = 0.3$ mg/mL) on silicon wafers (page S9 in SI). Moreover, dynamic light scattering (DLS) analysis confirmed the block copolymer could self-assemble at low concentration (0.3 mg/mL) of the samples in absorption/emission. The hydrodynamic diameter of the self-assembled architecture obtained from DLS was 33.7 nm, generally agree with the AFM observations. We have added a description in the manuscript (page 14, line 17-20): "Dynamic light scattering analyses revealed that the hydrodynamic diameter of poly(**2₂₀-b-1b₄₀**) was 33.7 nm in IPA, even at a low concentration of 0.3 mg/mL, which imply that the block copolymer was self-assembled in IPA

(Supplementary Fig. 32).”

(4) The authors state "with the presence of selective solvent IPA, the polymer self-assembled into compacted spherical nanoparticles."

What is the evidence for this? For example, can this be observed by DOSY NMR?

Response: We have tried DOSY NMR experiment, however the signals were very broad due to the self-assembly and could not obtain useful information. Thus, we determined the size of the self-assembled nanoparticles using dynamic light scattering (DLS) analysis, a method that commonly used in block copolymer self-assembly (*J. Am. Chem. Soc.* 2018, 17773–17781; *Macromolecules* 2016, 110–119; *Macromolecules* 2016, 1180–1190). DLS analysis revealed the hydrodynamic diameter of the block copolymer was ca. 33.7 nm in IPA, suggested it was self-assembled in IPA (page 14, line 17-20). Transmission electron microscopy (TEM) observation confirmed the block copolymer formed core-shell-like spherical nanoparticles with 32 nm diameter in IPA and with good homogeneity. We have added a description in the manuscript (page 14, line 11-16).

(5). In Table 1, entries 1-6 do not have M_n data. Why? Was it lower than the detection limit?

Response: Yes, the M_n was lower than the detection limit of SEC. We have revised footnotes of Table 1 and Supplementary Table 1 in SI to address this issue. Owing to the limit of the displayed items in main text, Table 2 in the original manuscript was moved to SI and was denoted as Supplementary Table 1 in the revised manuscript.

(6). The sentence beginning with "Nickel catalysts have been widely used in polymer synthesis..." needs some citations to examples of Nickel catalysts in polymer synthesis to support the authors statement that they have been widely used.

Response: The related papers about the utilization of nickel(II)-catalysts in polymer syntheses have been cited in ref. 30 and 31 (*Chem. Rev.* 2015, 12091–12137; *Chem. Rev.* 2016, 1950–1968) in the manuscript (page 3, line 5). We appreciate this reviewer for the important suggestions.

Reviewer #3:

Comments: This is an interesting paper that describes a novel living polymerization of diazoacetates with a non-expensive Ni(II) catalyst and further living block

polymerization of 3-hexylthiophene, resulting in photoluminescence block copolymers. Poly(acylmethylene)s, an interesting class of polymers that are different from commodity plastics, polyolefins, have been extensively studied during the past decade. Nowadays Pd(II)-based catalysts developed by the group of Wu and others enable to polymerize diazoacetates in a living fashion to produce the poly(acylmethylene)s with a controlled molar mass and a narrow molar mass distribution. The main advance in this paper lies in the discovery of a unique bisthieryl Ni(II) catalyst carrying a dppp ligand (BT-Ni(dppp)Cl), which promoted the living homo- and block polymerizations. The development of non-expensive, non-noble metal, Ni-based catalysts/initiators for living polymerization of diazoacetate appears to be a great achievement, and the results are worthy of publication.

The work has been carried out to a high standard and the manuscript is well written based on the experimental synthesis results. It is not clear, however, whether the work has the exceptional level of significance required for publication in this journal. Perhaps, the mechanism of the living polymerization of diazoacetates initiated and propagated by the BT-Ni(dppp)Cl should be proposed/discussed. The rationale for the reactivity of BT-Ni(dppp)Cl that is better than PhNi(dppp)Br should be also discussed; DFT calculations using crystal structures of BT-Ni(dppp)Cl and PhNi(dppp)Br may be useful. In addition, the authors should stress the superiority of the present photoluminescence block copolymers over other PT-based block copolymers together with physical properties of novel homo- and block copolymers. Otherwise, publication in a more specialized journal might be appropriate.

Response: Many thanks to this reviewer for the positive comments and valuable suggestions. We have revised the manuscript to address these issues: (1) Mechanism of the Ni(II)-catalyzed living polymerization of diazoacetate was proposed in the manuscript, and was supported by DFT calculations. We have attempted to isolate the catalyst crystal, however it was continually failed. Therefore, the DFT calculations were performed without using the crystal structures (see page S10 in SI, Supplementary Table 2, and Supplementary Fig. 33-Fig. 39 for details). Furthermore, the rationale for the reactivity of BT-Ni(dppp)Cl that is better than PhNi(dppp)Br was also discussed. We have added a discussion in the manuscript (page 15 and Fig. 8).

(2) The P3HT-*b*-polycarbene copolymer synthesized in this manuscript did have some superiorities. We have added an explanation in the main text (page 14, line 2-5 from the bottom): “It is noteworthy that the polycarbene block is composed of C–C bonds and bearing densely packed substituents on every backbone atom, which

renders it with relatively high rigidity and high self-assembly tendency, thereby imparting the P3HT-*b*-polycarbene copolymer with distinct self-assembly induced PL.^{18,21,24}

REVIEWERS' COMMENTS

Reviewer #1 (Remarks to the Author):

In my view the authors have properly addressed the points indicated by the reviewers. I think the paper can be published now.

Reviewer #2 (Remarks to the Author):

While the English still needs improvement, the authors have adequately addressed the reviewer concerns. No further experiments are required prior to publication.

Reviewer #3 (Remarks to the Author):

The authors performed additional series of experiments and calculations according to the comments and suggestions by reviewers and further modified and revised the manuscript and Supporting Information (SI) based on the new results. In my opinion, the revised version of the manuscript has been certainly improved and may be acceptable for publication in this journal except for the DFT calculation results as follows.

The proposed mechanism (Figure 8) of the living polymerization of diazoacetates initiated and propagated by the nickel catalysts (BT-Ni(dppp)Cl and PhNi(dppp)Br) seems to be acceptable in general. However, the following description (page 14-15) of the DFT calculation results (Table S2 and Figures S33 and S34) was difficult to follow and understand;

"Based on DFT analyses, the intermediate IIb formed more easily by Ph-Ni(dppp)Br and is more stable than the analog IIa formed by BT-Ni(dppp)Cl. Thus, the chain extension of the diazoacetate monomer by IIb is more difficult than that of IIa. Therefore, BT-Ni(dppp)Cl leads to the formation of anticipated polymers in high yield with predictable Mn, whereas Ph-Ni(dppp)Br only yields low molecular weight polymers in low yield under the same reaction conditions." IIa and IIb are correct?

The authors should discuss the mechanism of the living polymerization and difference in the polymerization activity between BT-Ni(dppp)Cl and PhNi(dppp)Br more properly in more details using the transition states (Ph-TS1 and Ph-TS2 versus BT-TS2) and ground states of the propagation species (Ph-IN4 versus BT-IN4) based on the DFT calculation results.

Both the catalysts catalyzed the living polymerization of 1a (Table 1, runs 12 and 14) in a different polymerization rate, therefore, in my opinion, the preliminary DFT calculation results with the lack of the crystal structures of the catalysts may not be convincing for further discussion of the difference in the polymerization activity between BT-Ni(dppp)Cl and PhNi(dppp)Br based on the difference in the stabilities of IIa and IIb and perhaps further calculations of IIIa and IIIb may be required, which, however, are not a condition for acceptance.

Reviewer #1:

Comment: In my view the authors have properly addressed the points indicated by the reviewers. I think the paper can be published now.

Response: we appreciate this reviewer for the positive comment.

Reviewer #2:

Comments: While the English still needs improvement, the authors have adequately addressed the reviewer concerns. No further experiments are required prior to publication.

Response: The English of this manuscript has been further improved by a native English-speaking expert.

Reviewer #3:

Comments: The authors performed additional series of experiments and calculations according to the comments and suggestions by reviewers and further modified and revised the manuscript and Supporting Information (SI) based on the new results. In my opinion, the revised version of the manuscript has been certainly improved and may be acceptable for publication in this journal except for the DFT calculation results as follows.

(1) The proposed mechanism (Figure 8) of the living polymerization of diazoacetates initiated and propagated by the nickel catalysts (BT-Ni(dppp)Cl and PhNi(dppp)Br) seems to be acceptable in general. However, the following description (page 14-15) of the DFT calculation results (Table S2 and Figures S33 and S34) was difficult to follow and understand; “Based on DFT analyses, the intermediate IIb formed more easily by Ph-Ni(dppp)Br and is more stable than the analog IIa formed by BT-Ni(dppp)Cl. Thus, the chain extension of the diazoacetate monomer by IIb is more difficult than that of IIa. Therefore, BT-Ni(dppp)Cl leads to the formation of anticipated polymers in high yield with predictable Mn, whereas Ph-Ni(dppp)Br only yields low molecular weight polymers in low yield under the same reaction conditions.” IIa and IIb are correct?

Response: The IIa and IIb were interchanged by a careless mistake, which was corrected in the revised manuscript. Moreover, more detailed description of DFT calculations has been added in the manuscript according to the reviewer’s suggestion (page 15, line 1-4 from the bottom; and page 16-17).

(2) The authors should discuss the mechanism of the living polymerization and

difference in the polymerization activity between BT-Ni(dppp)Cl and PhNi(dppp)Br more properly in more details using the transition states (Ph-TS1 and Ph-TS2 versus BT-TS2) and ground states of the propagation species (Ph-IN4 versus BT-IN4) based on the DFT calculation results.

Response: We have discussed the mechanism of the living polymerization and difference in the polymerization activity between BT-Ni(dppp)Cl and Ph-Ni(dppp)Cl based on the DFT calculation results. The description was added in the manuscript (page 15, line 1-4 from the bottom; and page 16-17).

(3) Both the catalysts catalyzed the living polymerization of 1a (Table 1, runs 12 and 14) in a different polymerization rate, therefore, in my opinion, the preliminary DFT calculation results with the lack of the crystal structures of the catalysts may not be convincing for further discussion of the difference in the polymerization activity between BT-Ni(dppp)Cl and PhNi(dppp)Br based on the difference in the stabilities of IIa and IIb and perhaps further calculations of IIIa and IIIb may be required, which, however, are not a condition for acceptance.

Response: We really appreciate the reviewer's suggestion. However, the influence of the Ph and BT units on the polymerization would decrease considerably with the chain extension because the distance between these units and the living chain end increases. We have added a description in the manuscript (page 16, line 1-2 from the bottom).

We appreciate the professional and important suggestions and comments from the reviewers. We hope that this revised manuscript with improved quality can meet the high standards of Nature Communications.